# Therapeutic Perspectives on ROCK Inhibition for Cerebral Cavernous Malformations

**Tadeu L. Montagnoli** [1,2,*] , **Daniela R. de Oliveira** [1,2] and **Carlos A. Manssour Fraga** [1,2,*]

1 Laboratório de Avaliação e Síntese de Substâncias Bioativas (LASSBio), Instituto de Ciências Biomédicas, Universidade Federal do Rio de Janeiro, Rio de Janeiro 21941-902, Brazil

2 Programa de Pós-Graduação em Farmacologia e Química Medicinal, Instituto de Ciências Biomédicas, Universidade Federal do Rio de Janeiro, Rio de Janeiro 21941-902, Brazil

* Correspondence: tmontagnoli@gmail.com (T.L.M.); cmfraga@ccsdecania.ufrj.br (C.A.M.F.)

**Abstract:** Cerebral cavernous malformations (CCM) are developmental venous dysplasias which present as abnormally dilated blood vessels occurring mainly in the brain. Alterations in vascular biology originate from somatic mutations in genes regulating angiogenesis and endothelial-to-mesenchymal transition. Vascular lesions may occur at any time and develop silently, remaining asymptomatic for years. However, symptomatic disease is often debilitating, and patients are prone to develop drug-resistant epilepsy and hemorrhages. There is no cure, and surgical treatment is recommended only for superficial lesions on cortical areas. The study of lesion biology led to the identification of different pathways related to disease onset and progression, of which RhoA/Rho-associated protein kinase (ROCK) shows activation in different subsets of patients. This work will explore the current knowledge about the involvement of ROCK in the many aspects of CCM disease, including isoform-specific actions, and delineate the recent development of ROCK inhibitors for CNS-targeted diseases.

**Keywords:** cerebral cavernous malformation; CCM; CNS; blood–brain barrier; ROCK; Rho-associated protein kinase; RhoA; kinase inhibitors





Cerebrovascular disease is a leading cause of mortality and disability worldwide, and among the estimated 12 million new cases in 2019, nearly half were fatal [1]. Vascular malformations are non-neoplastic anomalies which can appear in any part of the body, but their cerebral forms pose great risk of mortality, especially if hemorrhagic or neurological symptoms develop. Cerebrovascular anomalies are divided into four categories: arteriovenous malformations, which are shunting lesions linking arteries and veins; cerebral telangiectasias, predominantly constituted by anomalous capillaries; and cavernous malformations and developmental venous anomalies, comprising alterations of venous origin [2]. Cerebral cavernous malformations (CCM) are one of the most prevalent vascular anomalies in the central nervous system (CNS) and a significant cause of pediatric strokes [3–5]. Its pathogenesis remains partially understood and its treatment relies only on neurosurgical resection, if feasible. Therefore, the search for biological targets involved in disease biology is of great importance for the management of CCM. Recently, the role of the ras homolog family member A (RhoA)/Rho-associated coiled coil-containing protein kinase (ROCK) pathway has been described in the development of neurologic and vascular diseases, including those from cerebral vascular beds. However, no previous work thoroughly addresses the recent advances in CCM research with focus on this pathway. Therefore, this work summarizes the involvement of ROCK in CCM pathophysiology and critically reviews the recent development of ROCK inhibitors targeting CNS diseases.

## 1. Cerebral Cavernous Malformations (CCM): Epidemiology, Clinical Manifestations, and Current Therapy

Cerebral cavernomas, or CCM, are vascular dysplasias characterized by clusters of non-shunting abnormally dilated blood vessels ("caverns") with multilobulated mulberry-like

appearance, which present low blood flow and are prone to thrombosis and hemorrhage (Figure 1) [6–8]. These structures occur on the venous side of the capillary bed, which makes CCM angiographically occult and only detectable by magnetic resonance imaging (MRI) [6,8]. Despite being also named "hemangiomas" or "angiomas", CCM lesions display low proliferative activity and are classified as simple non-neoplastic venous malformations by the International Society for the Study of Vascular Anomalies (ISSVA) [2,6,9,10].

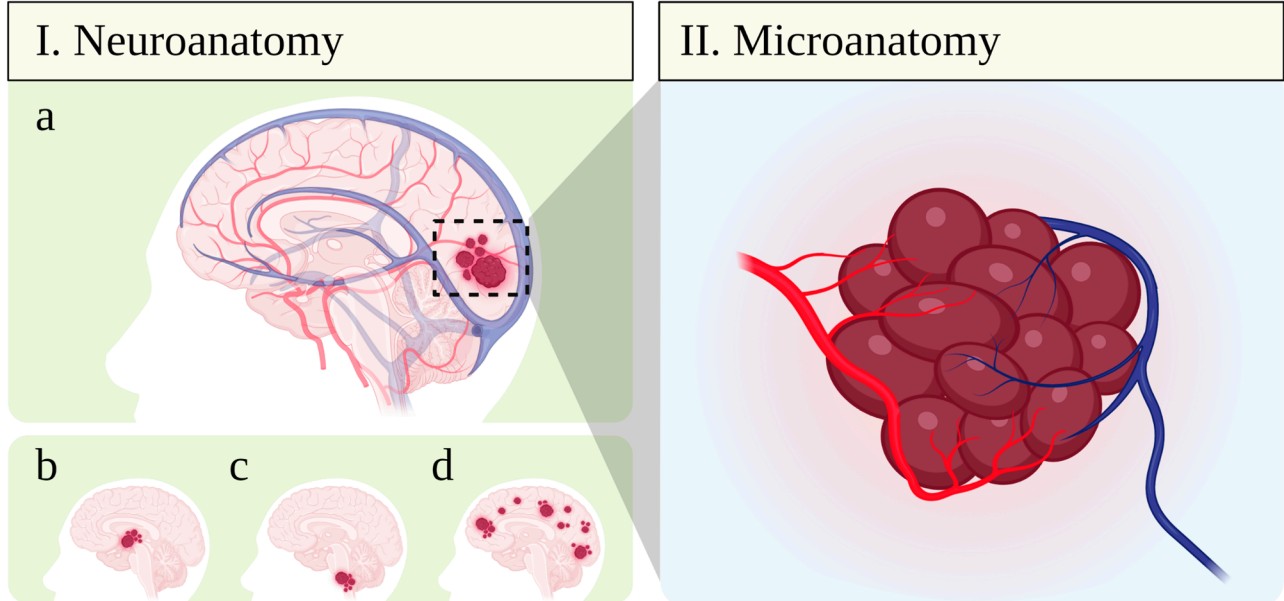

**Figure 1.** Neuroanatomical features of cerebral cavernous malformations. Commonly affected locations in the central nervous system include cortex (**Ia**), "deep seated" areas (**Ib**), and brainstem (**Ic**) and lesions may present as isolated or multiple (**Id**), depending on disease etiology. Lesion microanatomy appears as mulberry-like structures of ectatic post-capillary vessels (**II**).

Recent studies indicate a prevalence in 2–9 individuals per thousand and an incidence of 5–6 new diagnostics of CCM per million adults per year [11,12]. Prevalence shows a slow increase with age, although pediatric cases constitute 25–35% of the population living with CCM [4,5,9,11,12]. The disease is not restricted to any ethnic group, but some ethnicities (Hispanics and Asians) may display higher prevalence rates [6,11]. Moreover, biological sex [6,11,12], pregnancy [13,14], and cardiovascular risks [12,15,16] are not involved in disease progression or symptom development.

Cavernous malformations are acquired lesions which originate sporadically or in a familial pattern [6,8,11]. Familial CCM corresponds to approximately 20% of all affected individuals, which usually present multiple scattered lesions [6,8,11,17]. This form is associated with autosomal dominant inheritance of gene mutations affecting vascular biology and shows incomplete penetrance, i.e., only a fraction of carriers develop the disease [6,17]. On the contrary, sporadic CCM is found in up to 80% of patients and appears as an isolated lesion or multiple lesions circumscribed to a restricted area, often in proximity with developmental venous anomalies [6,8,17]. Its pathogenesis remains largely unknown, but genetic origins also seem to be implicated [6,8,11]. Moreover, CCM lesions may develop after radiation therapy or appear *de novo* during one's lifetime [6,9,11,18–20].

Although usually restricted to the CNS, CCM can also appear in the meninges, nerves, retina, and skin [6,21]. Within the CNS, 65–80% of lesions are found in supratentorial areas, while brainstem, cerebellar, and intraspinal lesions are less prevalent and commonly associated with the familial form [11,21–23]. With the advent of MRI, a remarkable increase in incidental detection of silent cavernomas has been reported. Symptoms associated with CCM appear in 10–80% of diagnosed cases, depending on the population studied [6,12,24].

Patients previously treated with radiotherapy are also at greater risk of developing symptomatic disease, due to a 6-fold increase in the prevalence of CCM in this population [11,20].

Neurovascular symptoms are often debilitating and sometimes life-threatening, ranging from headaches, muscle weakness, and chronic pain to seizures, neurological deficits, and hemorrhagic strokes. Giant cavernomas sized over 3 cm are seldom found and may cause a substantial compressive mass effect on surrounding areas [21,25,26]. Symptoms often manifest after the second decade of life, but can occur at any age [8,27]. Supratentorial CCM is commonly associated with seizures and usually evolve to epilepsy within 1 year, which is treated with anticonvulsant therapy or neurosurgery [11,18,25,28,29]. The incidence of seizures is higher in pediatric CCM patients [18], and surgical excision may prevent neurological defect expansion and psychosocial disabilities due to long-term anticonvulsants [5]. Due to anticonvulsant teratogenicity, surgery is also recommended to pregnant women with symptomatic CCM up to the second trimester or if severe symptoms develop [13,14].

Intracerebral hemorrhage is the most severe consequence of the disease, which may lead to fatal outcomes [4,11,30]. It is one of the main reasons for clinical presentation, whether from individuals bearing familial or sporadic CCM, as both demonstrate similar susceptibility [11,12]. Whenever possible, surgical resection is indicated for symptomatic CCM lesions, as the risk of bleeding increases from 0.5–2% to 4.5–23% annually after a previous hemorrhage [30]. Deep-seated and brainstem lesions are found in 10–20% of cases and are associated with greater risk of first (2.3–6.8% per year) and re-hemorrhage (21–62% per year) [11,30]. The vicinity of these lesions to eloquent structures causes great concern, often leading to rapid clinical deterioration and life-threatening outcomes after hemorrhage [11,30]. However, neurosurgical resection of these lesions is also a high-risk procedure and may cause postoperative hemorrhages and transient or permanent disabilities, not being generally recommended [11,17,30,31]. In these cases, the use of radiosurgery or laser ablation for CCM removal has been reported [9,11,17,32–34], but their efficacy remains uncertain.

## 2. Pathobiology of the Neurovascular Unit in CCM

The exact mechanism behind CCM pathogenesis remains uncertain, although aberrant vasculogenesis has been suggested. Lesions present ectatic blood vessels in a complex labyrinthic arrangement separated by loose connective tissue with but without intervening brain parenchyma (Figure 2) [35]. At the cellular level, all components of the neurovascular unit show distinct degrees of degeneration. An outer rim of reactive astrocytes is often encountered encircling the cavernoma [25,35], although neither neuronal or glial cell processes penetrate the lesions. Vessels are completely devoid of smooth muscle and pericytes are rarely seen. Their luminal surface is composed by a single layer of endothelial cells with defective intercellular junctions laying on a thin and discontinuous basal lamina [35,36]. Moreover, hemosiderin deposits indicating hemoglobin breakdown are commonly seen around the lesion [25,37]. These alterations indicate a defective blood–brain barrier (BBB) in CCM [36], which underlies their propensity to vasogenic edema and hemorrhage.

# 3. Lesion histopathology

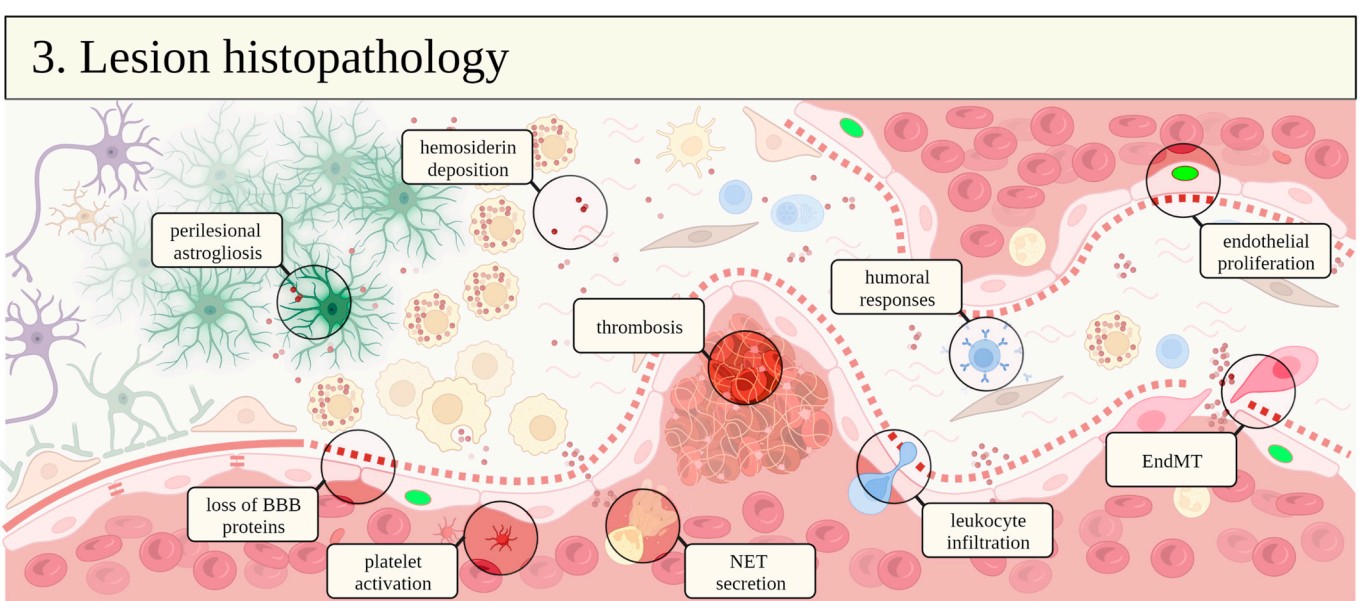

**Figure 2.** Histological panorama of cerebral cavernous malformations. Lesions appear microscopically as dilated congested capillaries with simple endothelium and discontinuous basal membrane. Endothelial cells proliferate and undergo endothelial-to-mesenchymal transition (EndMT), contributing to angiogenesis and extracellular matrix turnover. Loss of blood–brain barrier (BBB) proteins and cell junctions lead to plasma protein leakage and microhemorrhages. Hemosiderin, a byproduct of hemoglobin degradation, may be phagocytosed and cleared by macrophages or remain in the extracellular matrix, causing perilesional astrogliosis. Myeloid (yellow) and lymphoid cells (blue) also infiltrate lesions and participate in tissue remodeling, humoral responses, and thrombosis.

Under physiological conditions, nutrient supply to the CNS depends on transport across the BBB, due to its reduced permeability to paracellular and transcellular routes [38,39]. Its development and integrity results from neurovascular interactions in a series of time- and spatially regulated signaling pathways [38]. Barriergenesis is preceded by angiogenesis, in which new immature leaky capillaries are formed in response to trophic stimuli, mainly vascular endothelial growth factor (VEGF) [38]. In fact, CCM lesions display elevated protein levels of VEGF and higher angiogenic signaling [40,41], which stimulates the disruption of cell–cell and cell–extracellular matrix junctions for transdifferentiation of the quiescent endothelium into a highly proliferative and migratory phenotype [42]. This process of endothelial-to-mesenchymal transition (EndMT) is considered pivotal for the pathogenesis of CCM [43,44].

The first cues for elucidating its pathophysiology came from studies on familial CCM, which shows autosomal dominant inheritance. Between 2000 and 2005, loss-of-function germline mutations were identified in three genes, named *Krev Interaction Trapped 1 ankyrin repeat containing* (*KRIT1* or *CCM1*, HGNC:1573, OMIM:604214), *Cerebral Cavernous Malformation 2 scaffold protein* (*CCM2*, HGNC:21708, OMIM:607929), and *Programmed Cell Death 10* (*PDCD10* or *CCM3*, HGNC:8761, OMIM:609118), which contribute to roughly 60, 20, and 10% of the familial cases, respectively [11,45]. As germline mutations do not explain the CNS selectivity, the detection of a second somatic mutation in lesions from both familial and sporadic forms substantiated a "two-hit" hypothesis [8,11,46]. However, animal studies illustrated that *CCM1-3* mutations do not correlate with human disease onset or progression [46–48], and the search for a "third hit" pointed to a series of pathological processes, such as epigenetics [49–51], oxidative stress [31,52–54], inflammation [31,46–48], local hypoxia [47,55], defective autophagy [46,47,53], angiogenic signaling [31,46,47], EndMT [31,46,47,56], deranged hemodynamics [46,47,55,57], and thrombosis [55].

More recently, somatic gain-of-function mutations in genes *Mitogen-Activated Protein Kinase Kinase Kinase 3* (*MAP3K3*, HGNC:6855, OMIM:602539) and *Phosphatidylinositol-4,5-bisphosphate 3-Kinase, Catalytic subunit α* (*PIK3CA*, HGNC:8975, OMIM:171834) have also been identified in sporadic and familial CCM as triggers for pathogenesis and disease progression, respectively [48,58–60]. The activation of Mitogen-Activated Protein Kinase Kinase Kinase 3 (MEKK3), the product of the *MAP3K3* gene, is also observed in endothelial cells bearing loss-of-function mutations in CCM1-3 [56,61]. This enzyme initiates a phosphorylation cascade, which activates transcription factors Krüppel-like factor-2 (KLF2) and -4 (KLF4) and ultimately mediates the endothelial response to angiogenic stimuli [46,56,61]. Among the pathways under transcriptional control by KLF2/KLF4, RhoA and its effector ROCK regulate contractility, adhesion, migration, and proliferation in CCM1 knockdown endothelial cells by modulating the structure of actomyosin filaments [46,61,62]. Endothelial cells from CCM lesions of patients and animal models display higher activity of the RhoA/ROCK pathway, which has been associated with increased disease progression and incidence of bleedings [46,63,64].

Given the current lack of pharmacological treatment for reducing disease burden, this pathway became attractive as a target for attenuating CCM development and hemorrhages. The inhibition of RhoA by statins is currently under investigation in an ongoing Phase I/II clinical trial (NCT02603328) [65]. This approach is based on the inhibition of geranylgeranyl pyrophosphate synthesis in the mevalonate pathway, the primary target of the statins, blocking RhoA prenylation and its membrane anchoring for ROCK activation [66]. Although this "pleiotropic" action of statins is commonly associated with beneficial effects, it is worth noting that it is not their primary intended use. Dosage used on the AT CASH EPOC trial would begin at 80 mg, the highest dose available for the control of dyslipidemia. Therefore, the increased risk of side effects, especially hemorrhagic strokes [66–69], generates concerns regarding the continuous use of high-dose statins, which could outweigh their clinical utility for CCM. In addition, the inhibition of protein prenylation affects not only RhoA but all other membrane-anchored proteins [36,69]. As endothelial cell biology involves a fine-tuning of GTPase activities [70,71], widespread inhibition of protein prenylation could potentially counterbalance the intended positive effects of RhoA inhibition.

To circumvent the limitations of statins, the inhibition of ROCK could provide enough efficacy and selectivity for the treatment of CCM, since its activation acts as a convergence point of many signaling pathways involved in cardiovascular diseases. In fact, the kinase inhibition approach outperforms statins in reducing disease burden in experimental models of CCM [72,73], indicating the usefulness of this pharmacologic strategy.

## 3. The Role of Rho-Activated Coiled Coil-Containing Protein Kinases (ROCK) in CCM

The Rho-associated coiled coil protein kinases (ROCK) are the most studied effectors of the small GTPase RhoA, and exist as two isoforms: ROCK1 (ROKβ or p160$^{ROCK}$), encoded by gene *ROCK1* (HGNC:10251, OMIM:601702) first isolated from human platelet extracts [74]; and ROCK2 (ROKα or Rho-kinase), encoded by gene *ROCK2* (HGNC:10252, OMIM:604002) and identified in rat brain extracts [75]. Both ROCK isoforms display similar tridimensional structures, with a *N*-terminal kinase domain and a *C*-terminal pleckstrin homology domain joined by a coiled coil region, which contains the Rho-binding domain [76]. Proteins are found as dimers, and enzyme activity is independent of previous (auto)phosphorylation [77–79]. Isoform homology reaches 65% along the entire primary sequences and surpasses 90% inside the substrate binding site [76,80]. Although both isoforms are ubiquitous, ROCK1 predominates in the kidney, liver, and hematopoietic organs, while ROCK2 shows greater abundance in the heart and CNS [80,81]. Gene knockout studies indicate non-redundant activities performed by each isoform [82–84].

The activity of direct ROCK inhibitors has been evaluated in mouse models of familial CCM, developed by associating of *Ccm1-3* hemizygous deletion with sensitization to genetic instability caused by the knockout of *Tumor protein p53 pathway corepressor 1* (*Trp53*,

HGNC:43652, OMIM:191170) or *MutS Homolog 2* (*Msh2*, HGNC:7325, OMIM:609309). When administered orally for 4 months to *Ccm1*$^{+/-}$ *Msh2*$^{-/-}$, *Ccm2*$^{+/-}$ *Trp53*$^{-/-}$, and *Ccm3*$^{+/-}$ *Trp53*$^{-/-}$ mouse models, fasudil (100 mg/kg/day) increased survival and reduced the density and severity of cerebral lesions [72,73,85]. In addition, fasudil also attenuated the perivascular hemosiderin deposition, indicating a reduced tendency of microhemorrhages, and B lymphocyte infiltration, which demonstrates reduced vessel inflammation [72,73]. In the brain, fasudil attenuated the phosphorylation of ROCK substrates in both lesion endothelium and perivascular leukocytes [72,73]. Although it showed promising efficacy, fasudil is a weak ROCK inhibitor and does not show selectivity towards any ROCK isoform [86]. Therefore, further studies using *Rock1* and *Rock2* haploinsufficient mice were designed to elucidate the role of each isoform in familial CCM. These experiments indicated a higher beneficial effect of knocking down *Rock2* than *Rock1* on disease prevalence, lesion density, and iron deposition [87].

Despite the results obtained with haploinsufficient mice, experiments conducted with a mildly ROCK2-selective fasudil derivative, (*R*)-BA-1049 (100 mg/kg/day), resulted in minimal improvements in disease burden when compared to fasudil at the same oral dose and treatment scheme [87]. These outcomes indicate, at least, how incomplete the understanding of ROCK biology is in the physiopathology of CCM. Therefore, this and the next section will address the current knowledge on ROCK involvement in the many aspects of CCM, including isoform-specific actions, and the development of CNS-targeted inhibitors.

### 3.1. Endothelial Cell and Blood–Brain Barrier Function

A marked hyperactivation of ROCK is observed in the endothelium of both sporadic and familial CCM lesions, indicating its potential role as a therapeutic target [36]. The downregulation of CCM proteins is associated with increased ROCK activity in these cells, ultimately increasing BBB permeability [88]. At intercellular junctions, CCM proteins assemble in a macromolecular complex which inactivates RhoA by recruiting RhoGTPase-activating protein 29 (ARHGAP29) [45,62,89]. Physical interaction between CCM1 and CCM2 at the membrane stimulates RhoA degradation by SMAD-specific E3 ubiquitin protein ligase 1 Smurf1, further reducing RhoA/ROCK activation. The CCM complex can promote RhoA degradation by SMAD-specific E3 ubiquitin protein ligase 1 (SMURF1)-mediated ubiquitinylation [45] and inhibition of MEKK3/KLF-2/4 gene expression [18,46,61,90]. Moreover, it also blocks β1-integrin signaling through integrin cytoplasmic domain-associated protein-1 (ICAP-1), resulting in further RhoA/ROCK inhibition [62,89,91]. Finally, CCM3 protein can also engage in another complex, striatin-interacting phosphatase and kinase (STRIPAK), reducing ROCK activity independently from CCM1 and CCM2 [92].

As a consequence of the referred pathways, loss-of-function mutations in any of the *CCM* genes result in strong ROCK activation, leading to stress fiber formation and actomyosin contraction, which ultimately destabilizes cell–cell junctions [45,46,55,62,93]. In brain endothelial cells, the phosphorylation of occludin and claudin-5 by ROCK was reported and contributes to disassembling tight junctions [89]. The influence of the CCM protein complex on each ROCK isoform was recently addressed and revealed a predominance of ROCK1 on inducing transcellular permeability in *CCM*-depleted endothelial cell cultures [83].

The CCM complex also stimulates Notch signaling, promoting senescence and resistance to oxidative stress while inhibiting EndMT and angiogenesis [45]. In contrast, disassembling the complex stimulates β-catenin nuclear translocation and promotes the expression of EndMT genes [45,94]. In microvascular endothelial cells, EndMT pathways initiated by transforming growth factor (TGF)-β, SNAIL, and SLUG converge to ROCK activation and its inhibition may contribute to significantly reducing endothelial cell phenotype switching [80,95]. Additionally, CCM lesions display a mosaic pattern, which indicates lesion growth and also depends on recruiting wild-type endothelial cells after clonal expansion of somatic mutants [46,96,97]. Recently, ROCK isoforms were also at-

tributed different roles in lesion progression: while ROCK 1 dominates the expression of proteases for extracellular matrix turnover and invasion, ROCK2 preferentially regulates the chemoattraction of wild-type endothelial cells and leukocytes [84].

Although the influence of CCM proteins on ROCK activation seems pivotal for familial CCM, this kinase can also influence other aspects of cavernoma endothelial cell biology. In the luminal surface of CCM lesions, the accumulation of von Willebrand factor (vWF) is reported and involves RhoA/ROCK pathway [55]. In addition, intracavernous thrombosis also indicates the susceptibility of CCM endothelium to procoagulant proteases, which also stimulate BBB disruption by protease-activated receptor (PAR)/RhoA/ROCK signaling [55].

Vascular inflammation is also implicated in endothelial dysfunction, and alterations in gut microbiome suggested the influence of immune responses on disease course [98]. Activation of toll-like receptor 4 (TLR4) stimulates lesion progression and hemorrhage in rodent models of CCM, while its expression is correlated with increased disease burden in humans [47,55,99–101]. In fact, symptomatic CCM patients express higher TLR4 content in circulating lymphocytes [101], which indicates a potential predictive use of this biomarker for risk stratification. In the CNS, the activation of endothelial TLR4 receptors also causes the loss of tight junctions by means of RhoA/ROCK2 signaling [102,103].

Vascular lesions of CCM are also prone to blood stasis and disturbed flow, and the activation of endothelial ROCK2 by these factors may also contribute to increased BBB permeability, possibly by EndMT pathways [104–106]. Moreover, the use of ROCK inhibitors or a small-interfering RNA-targeting *ROCK2* also abolished hypoxic-stimulated proliferation of microvascular endothelial cells without compromising viability, indicating that ROCK may also control phenotypic changes under ischemia [107].

Despite both ROCK isoforms inducing the expression of leukocyte adhesion factors in endothelium, ROCK2 is reported to control endothelial cytoskeletal remodeling during diapedesis [108]. However, unselective ROCK inhibitors revert these alterations and preserve BBB integrity both in vitro and in vivo [72,73,109,110].

### 3.2. Leukocytes and Inflammation

Vessel inflammation is increasingly recognized as a major player in CCM pathology, and altered plasma cytokines are found in different cohorts of symptomatic and asymptomatic CCM patients and rodent models [55,100,111–113]. Inflammatory signals from the neurovascular unit have recently been pointed as driving elements of lesion growth and maturation, by attracting hematopoietic cells and stimulating thrombosis [55,114]. Advanced lesions show perivascular leukocytes within the intercavernous septa, of either myeloid (macrophages, neutrophils) or lymphoid origin (plasma cells, B and T lymphocytes) [55,101,114–116]. Therefore, immune cell activation is also implicated in CCM disease progression.

The activation of ROCK enhances leukocyte recruitment to the vasculature by modulating cytoskeleton remodeling, cell adhesion. and invasiveness [117–120], which are favored by BBB disruption. The inhibition of both ROCK isoforms has demonstrated anti-inflammatory effects on preclinical models of cardiovascular and autoimmune diseases by modulating leukocyte migration and activation [117,121,122]. However, the selective knockdown of ROCK1 in vivo enhances macrophage migration, indicating adhesion, and diapedesis are predominantly mediated by ROCK2 [122,123]. Increased ROCK content in circulating leukocytes is also a predictor of cardiovascular event risk, including hemorrhagic stroke, in cardiovascular diseases [124,125].

In addition to diapedesis, ROCK also regulates T cell activation, proliferation, and cytokine production [117,118,126,127]. In both human and rodent CCM, elevated numbers of activated T lymphocytes are found around lesions [116], further demonstrating the importance of immune responses in this disease. Moreover, symptomatic CCM patients display a higher $T_h17/T_{reg}$ ratio, indicating the predominance of a pro-inflammatory T cell phenotype in advanced disease [101]. Both ROCK isoforms induce biased responses

in T cells, as ROCK1 promotes their polarization towards $T_h2$ and ROCK2 to $T_h1/T_h17$ phenotypes [117,122,127,128]. The inhibition of ROCK2 also blocks Signal Transducer and Activator of Transcription (STAT)-1 and -3 and activates STAT-5, thus favoring $T_{reg}$ polarization over $T_h1/T_h17$, and locally controlling immune cell activation [129,130].

Disease progression also involves humoral responses, with the deposition of immunoglobulins around vascular lesions and the infiltration of B lymphocytes [100,115,116]. Although perivascular B cell density is proportional to endothelial ROCK activity, the role of ROCK on these cells remains poorly understood. Although RhoA is essential for B cell survival and development, this effect is not ROCK-dependent [126]. In contrast, ROCK1 is implicated in antigen internalization through B cell receptors [126], and enzyme inhibition can potentially inhibit autoimmune responses in the cavernoma milieu.

The enzymatic inhibition of ROCK impairs monocyte adhesion to endothelial cells and reduces their migration, proliferation, and differentiation [122,131]. Additionally, it also stimulates the phagocytosis of apoptotic cells by macrophages [132], which serves as a signal for polarization towards a pro-resolutive anti-inflammatory (M2) phenotype. However, ROCK influence on macrophage polarization seems context-dependent, probably by dependence on ROCK isoform, since ROCK1 promotes the inflammatory M1 while ROCK2 promotes the M2 phenotype in models of inflammation [122,133].

The accumulation of neutrophils and dendritic cells in the vicinity of CCM lesions was recently reported in a mouse model with brain endothelial cell-specific *CCM3* knockout [114], although their exact role in lesion development is currently unaddressed. Neutrophils contribute to thrombus formation inside vessels by releasing neutrophil extracellular traps (NET) [55], which are directly dependent on ROCK activation, as well as migratory activity, adhesion to endothelium, and phagocytosis [118,134–138].

### 3.3. Platelets and Thrombosis

Despite the incidence of thrombi inside caverns being long known in CCM pathology, only recently has the importance of platelets in disease development been investigated. Patients undergoing anticoagulant therapy are reported to develop intracranial hemorrhages at lower rates in different cohort studies [57,139,140]. The formation of intralesional thrombi is stimulated by an interplay of anti- and procoagulant factors secreted by endothelial cells, mainly due to low shear stress [55]. The role of ROCK on the activation and aggregation of human and rodent platelets in response to agonists or proteases is well known [141–143] and is mainly mediated by ROCK2, as indicated by platelet-specific knockout models [144]. Additionally, the pharmacological inhibition of ROCK reduces the activity of NADPH oxidase in activated human platelets [141], reducing superoxide synthesis and increasing nitric oxide bioavailability, which contributes to alleviating endothelial dysfunction inside lesions.

### 3.4. Activated Glia and Epileptogenesis

Seizures are the most common symptoms of CCM, but only 50% of patients become seizure-free after 1 year of pharmacotherapy [25,28]. Epileptogenesis is thought to originate in the outer rim of reactive astrocytes located around the lesion periphery [25,145], possibly by the metabolic regulation of excitatory neurotransmitter release or by reducing adenosine content in the area [25]. The effects of ROCK inhibition in neuron cell biology are known, with ROCK1 predominating in morphological processes and ROCK2 preferentially regulating synaptic plasticity [146]. Although our knowledge on the involvement of ROCK in epilepsy is still incipient, some reports indicate the usefulness of ROCK inhibitors on patients and experimental models of focal [147] and generalized seizures [148,149].

Defective BBB exposes surrounding astrocytes to pro-epileptogenic mediators, such as thrombin [150] or albumin [25]. Thrombin-stimulated astrocytes display a ROCK-dependent reduction in glutamate uptake [150], which contributes to increased excitotoxicity in neurons surrounding the CCM lesion. Recently, an increase in ROCK2 expression has been observed in the astrocytes of a rodent model of epilepsy and correlated with increased

astrocytic proliferation [149]. The inhibition of ROCK2 would not only control glutamate dysmetabolism but, additionally, also exert neuroprotection, as seen in neuron cultures exposed to kainate-induced excitotoxicity [151] and in hippocampal neurons of epileptic mice [152].

Although many aspects of CCM physiopathology have been recently elucidated, the contribution of the RhoA/ROCK pathway to this disease remains poorly understood. Its activation results from multiple signals, but the resulting effects are context-dependent and isoform-specific to each cell type involved. The inhibition of ROCK may not only slow lesion growth but also minimize the development of symptoms, such as seizures and hemorrhages, and thus can be considered a potential pharmacological target for CCM. Thus, the use of CNS-targeted ROCK inhibitors will provide further insights on its contribution to the disease and guide the discovery of new drugs for cavernoma management.

## 4. Recent Development of CNS-Targeted ROCK Inhibitors

Intracellular ROCK signaling plays a vital role in the morphogenesis and functions of the CNS, controlling processes such as axonogenesis, neuronal migration, and synaptic plasticity [153]. Since its kinase activity is enhanced in several neurologic and neurovascular pathologies, ROCK inhibition is a relevant strategy for treatment of these disorders and is shown to provide significant improvements in experimental models [146,154]. The development of new ROCK inhibitors for this purpose has attracted crescent interest of drug discovery companies, as indicates the number of applied patents worldwide between January 2003 and December 2022 (Table S1).

Retrieved patents displayed a constant annual rate of five new applications in the last 20 years, indicating the development of CNS-targeted ROCK inhibitors displaying continuous expansion (Figure 3A). In addition, this field has attracted continuous interest from pharmaceutical companies as, each year, nearly half of the applicants identified correspond to newcomers with a first patent in this field (Figure 3B,C). However, neurologic indications in these documents are rarely accompanied by evidence of adequate efficacy due to a lack of preclinical testing in experimental models. It must be emphasized that efficacy for treating CNS diseases is only obtained by proper balance of pharmacodynamic and pharmacokinetic properties [155], and is more precisely evaluated in animal models than directly in cell cultures. Efforts towards developing CCM-targeted ROCK inhibitors emerged more recently but also lack solid experimental data, with contributions from US companies BioAxone Biosciences and Cervello Therapeutics (Table S1).

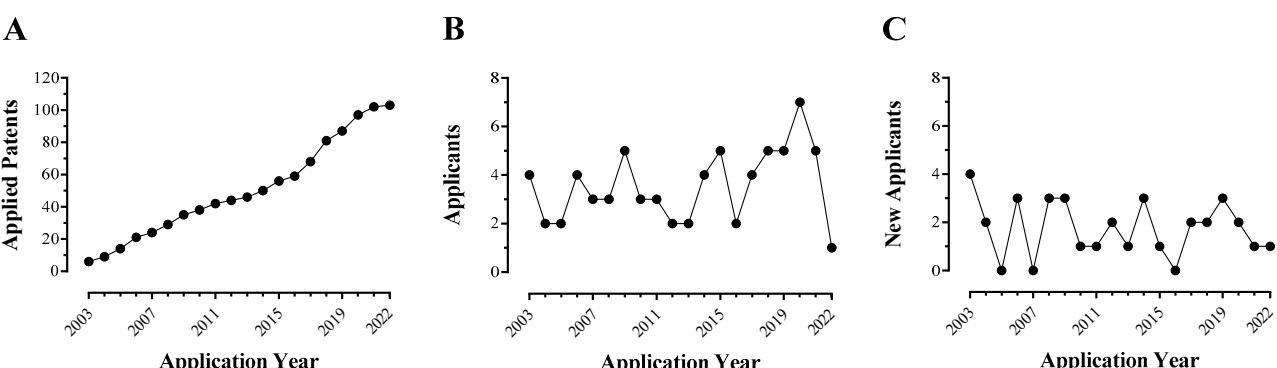

**Figure 3.** Analysis of published patents of ROCK inhibitors with indications for neurological diseases available from the Cortellis Drug Discovery Intelligence database between January 2003 and December 2022. (**A**) Cumulative time evolution of published patents. (**B**) Annual rate of applicant companies identified. (**C**) Annual rate of new applicant companies at time of first patent application.

In contrast to the analyzed patents, the scientific literature contains robust evidence of adequate efficacy in animal models, although the diversity of ROCK inhibitors remains highly unexplored. The examples found are presented in Table 1.

**Table 1.** ROCK inhibitors under preclinical development for neurological and neurovascular diseases.

| Compound | Affinity (pK$_D$) | | | Activity (pK$_I$) | | | BBB Score [155] | Detect. CNS? | CNS Testing | | |
|---|---|---|---|---|---|---|---|---|---|---|---|
| | ROCK1 | ROCK2 | Ref. | ROCK1 | ROCK2 | Ref. | | | Disease | Animal Model | Ref. |
| **Non-selective ROCK inhibitors (isoquinolines)** | | | | | | | | | | | |
| Fasudil (1)  | 6.04; 7.29 | 6.03; 7.34 | [86,156] | 6.46 | 7.02 | [157] | 4.90 | Yes [158] | fCCM | *Ccm1$^{+/-}$Msh2$^{-/-}$* mice | [72,85] |
| | | | | | | | | | fCCM | *Ccm2$^{+/-}$Trp53$^{-/-}$* mice | [72] |
| | | | | | | | | | AD | Wistar rats | [159,160] |
| | | | | | | | | | ALS | SOD1$^{G93A}$ mice | [161] |
| | | | | | | | | | Depression | Kv7.4$^{-/-}$ mice | [162] |
| | | | | | | | | | Epilepsy | GAERS rats | [148] |
| | | | | | | | | | Epilepsy | Sprague–Dawley rats | [147] |
| | | | | | | | | | I/R | Mongolian gerbils | [163] |
| | | | | | | | | | I/R | Sprague–Dawley rats | [164] |
| | | | | | | | | | I/R | CSE$^{-/-}$ mice | [165] |
| | | | | | | | | | PD | C57BL/6 mice | [166] |
| | | | | | | | | | PD | Sprague–Dawley rats | [167] |
| | | | | | | | | | Poison CI | Sprague–Dawley rats | [168] |
| | | | | | | | | | MS/AE | C57BL/6 mice | [169] |
| | | | | | | | | | MS/AE | SJL/J mice | [170] |
| | | | | | | | | | Schizophrenia | C57BL/6 mice | [158] |
| | | | | | | | | | SCI | Sprague–Dawley rats | [171] |
| Hydroxyfasudil (2)  | 6.27 | 5.87 | [86] | 7.13 | 7.09 | [172] | 4.43 | Yes [158] | Dementia | Fischer-344 rats | [173] |
| | | | | | | | | | Ischemia | Rats | [174] |
| | | | | | | | | | Ischemia | Sprague–Dawley rats | [175] |
| | | | | | | | | | I/R | Sprague–Dawley rats | [176] |
| | | | | | | | | | SAH | Sprague–Dawley rats | [177] |
| FSD-C10 (3)  | n.r. | n.r. | – | 5.94 * | 6.15 * | [178] | 5.28 | n.r. | AD | C57BL/6 mice | [179] |
| | | | | | | | | | MS/AE | C57BL/6 mice | [180] |

**Table 1.** *Cont.*

| Compound | Affinity (pK$_D$) | | | Activity (pK$_I$) | | | BBB Score [155] | Detect. CNS? | CNS Testing | | |
|---|---|---|---|---|---|---|---|---|---|---|---|
| | ROCK1 | ROCK2 | Ref. | ROCK1 | ROCK2 | Ref. | | | Disease | Animal Model | Ref. |
| (*R*)-BA-1049 (4) | 6.96 | 7.23 | [156] | 6.19; 5.47 | 6.49; 6.84 | [156] | 4.70 | Yes [156] | fCCM fCCM I/R | *Ccm1*$^{+/−}$*Msh2*$^{−/−}$ mice *Ccm3*$^{+/−}$*Msh2*$^{−/−}$ mice C57BL/6 mice | [87] [87] [181] |
| L-F001 (6) | n.r. | n.r. | – | 6.10 * | 5.98 * | [182] | 4.58 | Yes [182] | AD PD | C57BL/6 mice C57BL/6 mice | [183] [184] |
| Y-27632 (5) | n.r. | n.r. | – | 6.71 | 7.22 | [157] | 4.45 | n.r. | Trauma CI Depression Sespis CI SAH SCI Epilepsy | C57BL/6 mice C57BL/6 mice Wistar rats Sprague–Dawley rats Sprague–Dawley rats GAERS rats | [185] [186] [187] [177] [188] [148] |
| GSK269962 (7) | n.r. | n.r. | – | 8.80 ** | 8.40 ** | [146] | 0.59 | n.r. | Depression | Wistar rats | [189] |

**Table 1.** *Cont.*

| Compound | Affinity (pK$_D$) | | | Activity (pK$_I$) | | | BBB Score [155] | Detect. CNS? | CNS Testing | | |
|---|---|---|---|---|---|---|---|---|---|---|---|
| | ROCK1 | ROCK2 | Ref. | ROCK1 | ROCK2 | Ref. | | | Disease | Animal Model | Ref. |
| **ROCK2-selective inhibitors** | | | | | | | | | | | |
| Belumosudil (8) | 5.13 | 7.19 | [156] | 4.62 | 6.98 | [190] | 2.18 | Yes [191] | I/R Depression | C57BL/6 mice C57BL/6 mice | [191] [192] |

* Values estimated using Cheng–Prusoff equation. ** Values correspond to pCI$_{50}$. Abbreviations: AD, Alzheimer's disease; AE, autoimmune encephalomyelitis; ALS, amyotrophic lateral sclerosis; BBB, blood–brain barrier; CI, cerebral injury; CNS, central nervous system; fCCM, familial cerebral cavernous malformation; I/R, cerebral ischemia/reperfusion; MS, multiple sclerosis; n.r., not reported; PD, Parkinson's disease; SAH, subarachnoid hemorrhage; SCI, spinal cord injury.

Fasudil (**1**) is the prototypic ATP-competitive ROCK inhibitor of the isoquinoline sulfonamide type (Table 1), which is non-selective towards ROCK isoforms and is approved for the treatment of cerebral vasospasm in Japan [193]. This compound is the only one currently undergoing Phase II clinical studies for neurologic disorders, including amyotrophic lateral sclerosis (NCT03792490; NCT05218668), tauopathies (NCT04734379), and dementia (NCT04793659). In addition, **1** displayed beneficial effects in animal models of Alzheimer's disease [159,194], depression [162], acute ischemic stroke [163], cerebral ischemia [195], spinal cord injury [196], schizophrenia [158], multiple sclerosis [170], and epilepsy [147] (Table 1). Furthermore, studies with **1** were also performed in murine models of CCM [72,85], demonstrating significant reduction in lesion burden, inflammation, and hemorrhage.

Despite being brain-penetrant and displaying efficacy in neurologic disease models, pharmacologic effects observed after an oral dose of **1** are attributed to its metabolite, hydroxyfasudil (**2**), generated by first-pass hepatic metabolism [197]. Compound **2** has similar affinity and BBB permeability to **1**, but displays improved inhibitory activity and kinome selectivity towards ROCK, although it is still not isoform-selective (Table 1). In addition, **2** also has improved pharmacokinetics when compared to **1**, including a longer half-life and oral bioavailability [158]. Treatment with **2** provided improvements in rat models of age- or neurodegenerative-related memory dysfunctions [173], cerebral ischemia [175,176], and stroke [174].

Structural analogs of **1** are also under investigation in experimental models of CNS disorders. FSD-C10 (**3**) was designed by bioisosteric substitution in **1**, resulting in the replacement of homopiperazine by pyrrolidine, and by the introduction of a ramification in ring position 2, previously reported to enhance inhibitory activity [198]. Its higher lipophilicity enhanced the predicted brain permeability compared to **1**, although its pharmacokinetics in CNS tissues remains unknown. Its efficacy was investigated in rodent models of multiple sclerosis [178], Alzheimer's disease [179], and neuroinflammation [180] and, even though beneficial effects were observed, it does not outperform **1**.

The molecular hybridization of **1** and Y-27632 (**5**) generated compound (*R*)-BA-1049 (**4**), which reduces CCM lesion burden, inflammation, and microhemorrhages in sensitized mouse models [87]. It displays similar affinity for both ROCK isoforms and mild ROCK2-selective activity, similar to **5** (Table 1). A drop in lipophilicity due to the substitution of primary amine for the secondary amino group of **1** reduced the predicted CNS permeability of **4** (Table 1) and reflected a lower peak concentration in mouse brains after intravenous administration (5.5–110 ng/g tissue for 5 mg/kg of **4** vs. 250 ng/g tissue for 10 mg/kg of **1**) [156,158]. Moreover, **4** is also metabolized to an active 1-isoquinolone metabolite, which is far less brain-penetrant than **2**, suggesting the existence of a threshold of polar group density or polar surface area for BBB permeation of isoquinolonesulfonamides. These differences in pharmacokinetics may explain the similar in vivo pharmacologic profile between **4** and **1,** despite the improved inhibitory activity of the first [72,73,85].

As CNS diseases, such as CCM, are often multifactorial in origin, the use of a multitarget approach has been widely explored in CNS drug development. The molecular hybridization of **1** and antioxidant α-lipoic acid led to compound L-F001 (**6**) [182], which was also evaluated for Alzheimer's [183] and Parkinson's diseases [184] (Table 1), demonstrating antioxidant and anti-inflammatory effects in CNS cells. The increased lipophilicity of **6** improved its brain penetration 5-fold relative to **1,** but did not significantly modify its elimination kinetics from rat plasma, as indicated by elimination half-life ($t_{1/2}$): 1.3 h for **6** (30 mg/kg p.o.) *versus* 2.34 h for **1** (6 mg/kg p.o.) [182,199].

Two additional cases of pan-ROCK inhibitors with distinct hinge-binding motifs other than isoquinoline were also tested in animal models with neurologic disturbances. Y-27632 (**5**), a 4-aminopyridine derivative, presented beneficial effects in experimental models of spinal cord and cerebral injuries [185,188] and cognitive impairments [187] (Table 1). Its BBB Score (4.45) was comparable to that of **1** (4.90), suggesting adequate CNS penetration, while its elimination kinetics in mice ($t_{1/2}$ 1.67 h at 1 mg/kg i.v.) were superior to **1** ($t_{1/2}$ 0.02 h at

1 mg/kg i.v.) [200], probably because of the distinct metabolic routes involved. However, in rat models of epilepsy [148] and brain injury [177], the intraperitoneal administration of **5** showed beneficial actions comparable to **1**, indicating such pharmacokinetic differences are species-dependent.

To date, the aminofurazan compound GSK269962 (**7**) is the most potent ROCK inhibitor tested in neurological conditions (Table 1). Compound **7** reduced depressive behaviors in corticosterone-injected female rats after intra-arterial administration [189], although its very low BBB Score (<4) suggests poor brain penetration. Whether this reflects ROCK inhibition by an active metabolite or direct action of **7** in CNS areas lacking a BBB, i.e., pituitary gland or area postrema, remains to be investigated.

Although a predominance of ROCK2 compared to ROCK1 was initially reported in brain tissues [201], recent data demonstrate that both isoforms are relevant to CNS diseases [202–204], since they perform non-redundant functions [82–84]. Belumosudil (**8**) is the only selective ROCK2 inhibitor approved for clinical use in graft-*versus*-host disease and is currently investigated in clinical trials of inflammatory conditions [205]. Despite no clinical trials in neurological disorders being reported, evidence from rodent models support a neuroprotective effect of **8** in stroke [191,206] and brain injury [207], which parallels non-selective ROCK inhibitors (Table 1). However, **7**′s lower BBB Score indicates that it hardly permeates the BBB and, therefore, its mechanism of action on the CNS is not completely elucidated. Although these data support a prominent ROCK2 activation in these conditions, it is not sufficient for excluding the contribution of ROCK1 to disease onset or progression. Moreover, it is not known if ROCK1 could substitute for ROCK2 after long-term isoform-selective inhibition. Carefully designed molecular biology studies are currently beginning to untangle the biological role of each isoform in neurovascular diseases, and the development of selective pharmacological probes must support these investigations. Finally, comparative studies with isoform-selective BBB-permeant inhibitors are of utmost importance for addressing these issues in vivo in models of CNS diseases.

## 5. Concluding Remarks

The management of CCM has greatly evolved over the last decades. Despite the advances towards understanding the mechanisms of pathogenesis and progression, this disease remains associated with great burden and risks of mild to severe neurological deficits. More importantly, mortality is correlated with intracerebral hemorrhages, which cannot be avoided but must be simply treated as CCM lesions become active. Today, the only therapeutic option for treatment is neurosurgical resection, but its feasibility depends on lesion location, size, and frequency of hemorrhage. Moreover, neurosurgical procedures are not devoid of adverse outcomes, with risks of transient to permanent neurological impairment and mortality.

Currently, there is no consensus or guideline for the pharmacological treatment of CCM, and clinical management is restricted to conservative therapy or derives from anecdotal off-label use of cardiovascular drugs. Despite being used for treating severe infant hemangiomas and showing positive effects in case reports and retrospective studies in CCM [57,208–210], nonselective β-blocker propranolol did not alleviate the frequency of intracerebral hemorrhages in a recent prospective randomized Phase II clinical trial (NCT03589014) [211]. The use of statins as inhibitors of the RhoA/ROCK pathway is also reported to be beneficial for treating CCM, although simvastatin did not improve hemorrhage rates but instead increased white matter permeability in unaffected brain areas [68]. A randomized Phase I/II clinical trial of high-dose atorvastatin in CCM is expected to be completed by 2024 (NCT02603328) and will ascertain its utility for managing hemorrhage outcomes [65].

Not only is the knowledge about CCM pathogenesis and clinical evolution incomplete, but the lack of validated preclinical models for testing new drug candidates also contributes to disparities in translational research. Despite its low cost and easier management, *Ccm*-mutated zebrafish proved insufficient for studying possible treatments, as these animals do

not develop cerebrovascular disease phenotypes [36]. As *Ccm1-3* knockout and knockdown strategies did not produce viable models in mice, sensitized, conditional, and inducible genetic models were developed for studying familial CCM, all of them with advantages and shortcomings [36]. Additionally, currently, no model of sporadic CCM has been reported. As this disease displays low prevalence, preclinical model development and validation should be addressed as soon as possible for minimizing patient enrollment and exposure to inefficacious drug candidates.

The utility of ROCK inhibitors has been evaluated in mouse models of familial CCM, reducing lesion density and dimensions and their tendency to bleeding [72,73,85,87]. However, the scientific literature still lacks a definite mechanism for the beneficial effects observed, including essential issues as the distinct roles of ROCK isoform in the cell types of the neurovascular unit. Medicinal chemistry work on developing BBB-permeant isoform-selective ROCK inhibitors has greatly advanced since the approval of fasudil [146,212], and a recent contribution by our group identified new ROCK inhibitors based on the sulphonylhydrazone motif, such as LASSBio-2065 [212]. The structural optimization of this compound for achieving ROCK2 selectivity and improved pharmacokinetics is currently under investigation and must prove essential not only for providing inhibitor probes for addressing mechanistic gaps but also to develop new treatments for this disease.

## 6. Data Collection and Analysis

The data search and collection on CCM clinical features and pathobiology (Section 2) were performed on the PubMed database, using the keywords "cavernous malformation", "cavernoma", and "CCM", using "CNS", "nervous", "brain", and "cerebral" as filters and restricting our analysis to documents published between 2010 and 2022. Data about the role of ROCK in neurovascular tissues and cells (Section 3) were obtained from both PubMed and Scopus databases by searching "ROCK" or "Rho kinase" in both title or abstract and applying filters for each cell type (endothelium, leukocytes, platelets, and glial cells). Those searches with more than 50 results were further filtered with CNS-related terms and publishing year between 2010 and 2022.

Articles and patents on ROCK inhibitors (Section 4) were retrieved from the Cortellis Drug Discovery Intelligence (CDDI) database (Clarivate, https://www.cortellis.com/drugdiscovery/home, accessed on 3 December 2022) by searching "ROCK" and "Rho-Associated Protein Kinase" and filtering results by assigned targets, including "Rho associated coiled-coil containing protein kinase 1 (ROCK1)", "Rho associated coiled-coil containing protein kinase 2 (ROCK2)", or "Rho kinase (ROCK) (nonspecified subtype)". The inhibitor search was further refined by filtering assigned conditions with CNS disease-related available terms, i.e., "Neurological disorders", "Cerebrovascular disorders", "Parkinson disease", etc. Finally, only those inhibitors with available inhibitory activity on both isoforms and reported action in vivo in animal models were selected for analysis. Data on affinity and activity were taken only from documents reporting paired experiments in both ROCK isoforms, and inhibitors were classified as isoform selective if $\Delta pK_I \geq 2.00$. BBB Scores were calculated as described elsewhere [155]. Preclinical testing for each inhibitor was obtained in an "Experimental pharmacology" search in CDDI. Similarly, patent documents were collected from CDDI and manually examined for clinical indications. Only those patents or patent applications published in 2003–2022 and containing indications for CNS diseases were included in our analysis.

**Supplementary Materials:** The following supporting information can be downloaded at: https://www.mdpi.com/article/10.3390/kinasesphosphatases1010006/s1, Table S1: Patents with CNS-targeted ROCK inhibitors (2003–2022).

**Author Contributions:** Conceptualization: T.L.M. and C.A.M.F.; Methodology, Investigation, Formal Analysis, Data curation, Writing—Original Draft Preparation: T.L.M. and D.R.d.O.; Writing—Review & Editing, T.L.M., D.R.d.O. and C.A.M.F.; Funding Acquisition, C.A.M.F. All authors have read and agreed to the published version of the manuscript.

**Funding:** This study was financed in part by the Coordenação de Aperfeiçoamento de Pessoal de Nível Superior—Brasil (CAPES)—Finance Code 001. The authors would like to thank INCT-INOFAR (BR, grant number 465.249/2014-0), Fundação Carlos Chagas Filho de Amparo à Pesquisa do Estado do Rio de Janeiro (FAPERJ, grant numbers E-26/010.001273/2016 and SEI-260003/003613/2022, fellowship to CAMF), Conselho Nacional de Desenvolvimento Científico e Tecnológico (CNPq, grant number 304394/2017-3), and Departamento de Ciência e Tecnologia (DECIT, Ministry of Heath, Brazil).

**Institutional Review Board Statement:** Not applicable.

**Informed Consent Statement:** Not applicable.

**Data Availability Statement:** All data used for analysis of patents is available as supplementary material to this paper.

**Acknowledgments:** Figures created with BioRender.com (Figure 1, license agreement ET251NEYKT for T.L.M.; Figure 2, license agreement JI251NET1F for T.L.M.).

**Conflicts of Interest:** The authors declare no conflict of interest.

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
