# Peer review of "Therapeutic Perspectives on ROCK Inhibition for Cerebral Cavernous Malformations"

_2813-3757, doi:10.3390/kinasesphosphatases1010006_

Round 1
Reviewer 1 Report
The authors proposed a comprehensive review of a cardiovascular alteration (cerebral cavernous malformations) involving RhoA/Rho associated protein kinase (ROCK) signaling.
The authors frame the disorder from an epidemiological and genetic point of view. They define the pathophysiology of the malformation and the neuronal repercussions with epileptic responses. Finally, the authors discuss the most recent molecules used in therapy.
The manuscript is clear and very linear. Furthermore, the figures are very explanatory of cardiovascular disease.
I have only one curiosity: since epilepsy is a secondary response associated with endothelial malformation, are there data in the literature on the possible therapeutic action of cannabidiol?
Author Response
The authors proposed a comprehensive review of a cardiovascular alteration (cerebral cavernous malformations) involving RhoA/Rho associated protein kinase (ROCK) signaling.
The authors frame the disorder from an epidemiological and genetic point of view. They define the pathophysiology of the malformation and the neuronal repercussions with epileptic responses. Finally, the authors discuss the most recent molecules used in therapy.
The manuscript is clear and very linear. Furthermore, the figures are very explanatory of cardiovascular disease.
I have only one curiosity: since epilepsy is a secondary response associated with endothelial malformation, are there data in the literature on the possible therapeutic action of cannabidiol?
Answer to Reviewer #1:
A search was carried out in the scientific literature to correlate a possible use of cannabidiol (and possibly other cannabinoids) for treatment of cerebral cavernous malformation (CCM) or their associated seizures, and we have not found any studies, despite cannabidiol being widely used for treating seizures in both humans and animal models. Due to its lower prevalence, CCM is considered a rare condition and, therefore, there is still much to be explored to expand the knowledge of the scientific community about this disease.
Reviewer 2 Report
The context should be better described in Introduction and the novelty character of this Review respect to the other ones in literature should be better marked.
Lines 321-331 should be better explained.
Lines 389-398 should be better written.
Figure 3 should be better described in the text.
Table 2 should be better discussed in the text.
A section methodologies , describing the criteria of bibliographic research should be inserted.
Author Response
POINT #1: The context should be better described in Introduction and the novelty character of this Review respect to the other ones in literature should be better marked.
Answer: We thank the suggestion. The introduction was partially changed in order to clarify the relevance and novelty of this review.
POINT #2: Lines 321-331 should be better explained.
Answer: We thank the suggestion. For better clarification, the paragraph was rewritten as follows:
“In addition to diapedesis, ROCK also regulates T cell activation, proliferation and cytokine production [118,119,127,128]. In both human and rodent CCM, elevated numbers of activated T lymphocytes are found around lesions [117], further demonstrating the importance of immune responses in this disease. Moreover, symptomatic CCM patients display higher Th17/Treg ratio, indicating the predominance of a pro-inflammatory T cell phenotype in advanced disease [102]. Both ROCK isoforms induce biased-responses in T cells, as ROCK1 promotes their polarization towards Th2 and ROCK2 to Th1/Th17 pheno-types [118,123,128,129]. Inhibition of ROCK2 also blocks Signal Transducer and Activator of Transcription (STAT)-1 and -3 and activates STAT-5, thus favoring Treg polarization over Th1/Th17 and locally controlling immune cell activation [130,131].”
POINT #3: Lines 389-398 should be better written.
Answer: The referred paragraph was rewritten as follows:
“Although many aspects of CCM physiopathology have been recently elucidated, the contribution of RhoA/ROCK pathway to this disease remains poorly understood. Its acti-vation results from multiple signals, but the resulting effects are context-dependent and isoform-specific to each cell type involved. Inhibition of ROCK may not only slow lesion growth but also minimize the development of symptoms, such as seizures and hemor-rhages, and thus can be considered a potential pharmacological target for CCM. Thus, the use of CNS-targeted ROCK inhibitors will provide further insights on its contribution to the disease and guide the discovery of new drugs for cavernoma management.”
POINT #4: Figure 3 should be better described in the text.
Answer: The text corresponding to Figure 3 was expanded for clarification and its reference to the presented graphs is indicated in the text, as follows:
“Retrieved patents displayed a constant annual rate of 5 new applications in the last 20 years, indicating the development of CNS-targeted ROCK inhibitors displays continuous expansion (Figure 3A). In addition, this field has attracted continuous interest of pharmaceutical companies as, each year, nearly half of the applicants identified correspond to newcomers with a first patent in this field (Figures 3B,C).”
POINT #5: Table 2 should be better discussed in the text.
Answer: We thank the observation and table 1 (which was erroneously numbered as table 2), was thoroughly cited and discussed in Section 4.
POINT #6: A section methodologies , describing the criteria of bibliographic research should be inserted.
Answer: As suggested, a description of the used search methodology was inserted at the end of the manuscript (Section 6).
Reviewer 3 Report
Dear authors,
This is a review about the cerebral cavernous malformations and the potential therapeutic role of ROCK inhibitors. The review is a significant contribution to the field, so no review on this topic has been previously published. The manuscript is well structured and written, and the appropriate references are included.
However, minor changes are necessary before publication. Specifically:
Lines 36-37: Why the authors write these name in italic letters? Italic letter should used only for name of genes.
Lines 44-48: If two sentences correspond to the same references, the authors could unify it.
Lines 56 and 59: change “panel” to figure”.
Line 76: “de novo” should write in italics.
Lines 145-146: the Morrison and Akers papers show variants of KRIT and PDCD10 genes related with CCM. The authors should include this information.
Line 154 and other: add the full name of genes.
Finally, a review to native speakers could improve the quality of the manuscript.
Author Response
This is a review about the cerebral cavernous malformations and the potential therapeutic role of ROCK inhibitors. The review is a significant contribution to the field, so no review on this topic has been previously published. The manuscript is well structured and written, and the appropriate references are included.
However, minor changes are necessary before publication. Specifically:
POINT #1: Lines 36-37: Why the authors write these name in italic letters? Italic letter should used only for name of genes.
Answer: We thank you for the observation. Italics here were intended to emphasize RhoA/ROCK as important players in CCM pathology, but we understand it may confound readers as referred genes are also in italics. We removed italics as suggested.
POINT #2: Lines 44-48: If two sentences correspond to the same references, the authors could unify it.
Answer: The referred sentences were elaborated from refs 6, 7 and 8, for the first one, and refs 6 and 8, for the second one, and describe the anatomical features of CCM lesions. We understand that merging both sentences would not provide a full description from anatomical point or could result in a longer unified sentence that would not be fully understandable. As the other reviewers did not make observations on this sentence, we choose to maintain the text as in the first version of the manuscript.
POINT #3: Lines 56 and 59: change “panel” to figure”.
Answer: We thank the suggestion and “figure” was substituted for “panel” in figure legends.
POINT #4: Line 76: “de novo” should write in italics.
Answer: We thank you for the observation. This typo was corrected as suggested.
POINT #5: Lines 145-146: the Morrison and Akers papers show variants of KRIT and PDCD10 genes related with CCM. The authors should include this information.
Answer: The authors understand the relevance of CCM1 and CCM3 gene variants for CCM pathology, as more than 50 mutations have been described in the literature for the first, as the Common Hispanic Mutation, and more than 30 for the latter. However, despite their importance for disease onset and development, we understand that further description does not add to the current manuscript, as most of these inactivating mutations lead to activation of Rho/ROCK pathway. Moreover, as indicated, further details on CCM gene variants are described in the reference works cited.
POINT #6: Line 154 and other: add the full name of genes.
Answer: The full names of genes and proteins were included in the revised manuscript as well as their entries on Human Genome Organisation Gene Nomenclature Committee (HGNC) and Online Mendelian Inheritance in Man (OMIM).
POINT #7: Finally, a review to native speakers could improve the quality of the manuscript.
Answer: We thank the suggestion and the present revised manuscript was submitted to an native English-speaker for improvements.